# Effects of Testing Methods and Sample Configuration on the Flexural Properties of Extruded Polystyrene

**DOI:** 10.3390/polym16131857

**Published:** 2024-06-28

**Authors:** Hiroshi Yoshihara, Masahiro Yoshinobu, Makoto Maruta

**Affiliations:** 1Faculty of Science and Engineering, Shimane University, Matsue 690-8504, Japan; yosinobu@riko.shimane-u.ac.jp; 2Faculty of Science and Technology, Shizuoka Institute of Science and Technology, Fukuroi 437-8555, Japan; maruta.makoto@sist.ac.jp

**Keywords:** extruded polystyrene, three-point bending test, four-point bending test, compression bending test, Young’s modulus, proportional limit stress, bending strength

## Abstract

Extruded polystyrene (XPS) is frequently used in the construction of many different structures. Therefore, it is necessary to appropriately characterize its mechanical properties to ensure the safety of said structures. Among the available characterization tests, static bending tests are simple and easy to perform; owing to these characteristics, they should be performed more frequently than other tests. In static bending tests on XPS, there are several challenges owing to the high flexibility of XPS, and the chosen testing method and sample configuration affect the accuracy of characterization. For cellular plastics, including XPS, three-point bending (TPB) test methods are standardized by the International Organization for Standardization (ISO) and Japanese Industrial Standards (JIS) as in ISO 1209-2:2007 and JIS K 7221-2:2006, respectively, where the sample configurations are determined. Therefore, TPB tests of cellular plastics have been conventionally performed based on these standardized methods to characterize the bending properties. In contrast, investigations on the effects of testing methods and sample configurations have often been neglected due to the existence of these standardized methods. However, to characterize the bending properties of XPS accurately, the effects of the testing method and sample configuration must be examined in detail. In this study, three bending properties (Young’s modulus, proportional limit stress, and bending strength) of samples cut from an XPS panel were determined using three-point bending (TPB), four-point bending (FPB), and compression bending (CB) tests with varying sample span/depth ratios from 5 to 50 at intervals of 5, and statistical analyses were performed to determine the relevance of the tests. The effect of sample configuration on Young’s modulus could be reduced when the span/depth ratio range was 25–50, 25–50, and 15–50 in the TPB, FPB, and CB tests, respectively, whereas that on the proportional limit stress was reduced in the span/depth ratio range of 5–50, 20–50, and 15–50 in the TPB, FPB, and CB tests, respectively. Additionally, the effect on the bending strength was reduced when the span/depth ratio range was 5–50, 20–50, and 5–50 in the TPB, FPB, and CB tests, respectively. Therefore, these results suggest that the TPB and CB tests were more feasible than the FPB test when the span/depth ratio was determined as being 25–50 and 15–50, respectively. However, clear differences were observed in the sample bending properties determined in these tests. In light of these findings, further studies should be conducted to elucidate these differences.

## 1. Introduction

Extruded polystyrene (XPS) possesses excellent thermal insulation properties; owing to these properties, it is frequently used as an insulating heat source in many different structures [1,2,3,4,5]. Of late, XPS panels have been used for floorings [6,7,8] and as sandwich panels [9,10,11,12] because their lightweight nature attenuates seismic forces. Additionally, the exceptional processability of XPS enhances its structural utilization. Therefore, adequate characterization of the mechanical properties of XPS is necessary to ensure the safety of structures in which XPS is used as the structural element.

A number of mechanical tests have been utilized to characterize the mechanical properties of foam materials, including XPS, such as static compression [4,13,14,15,16], static tension [15], flexural vibration [16,17], static shear [13,17,18], torsional vibration [18], asymmetric four-point bending [18], and static bending tests [14,15,16]. Among these tests, the static bending test is simple and easy to perform; therefore, to characterize the mechanical properties of XPS, this particular test should be used more frequently. In addition, three-point bending (TPB) tests of cellular plastics are standardized in ISO 1209-2:2007 [19] and JIS K 7221-2:2006 [20]; therefore, these standards have resulted in the increased utilization of TPB tests [21,22,23]. In addition, the four-point bending (FPB) test is used as frequently as TPB tests. However, at present, there are a number of concerns regarding the measurement of the mechanical properties of XPS, such as Young’s modulus, proportional limit stress, and bending strength. When a sample is not sufficiently slender, the deflection induced by the shearing force affects the measurement of the Young’s modulus and the proportional limit stress. In contrast, when a sample is extremely slender, the load–deflection behavior in the large deflection range deviates from that derived using elementary beam theory (EBT) because of the significant flexibility of XPS, and the bending strength cannot be predicted appropriately using equations based on EBT. Therefore, to appropriately characterize the bending properties of XPS using the aforementioned tests, the effect of the sample configuration must be considered in more detail. Independent of the TPB and FPB tests, compression bending (CB) tests have been performed using numerous carbon-fiber-reinforced plastics (CFRPs) [24,25,26]. In the CB test, bending deformation is induced by applying an axial load to the sample, and the load–deflection relationship can be determined based on elastica theory. Because the CB test can be performed without applying a lateral load to the sample body, there is no stress concentration due to the load, and bending failure due to the stress concentration can be avoided. CB tests involving solid wood [27], plywood and medium-density fiberboard (MDF) [28,29], and cardboard samples [30] at various span/depth ratios have been performed in previous studies, and the bending properties of these samples could be determined appropriately while reducing the effect of the sample configuration in the relevant span/depth ratio range. If the CB test, as well as the TPB and FPB tests, is applicable to measure the bending properties of XPS, then it shows great promise in the field. However, as described above, TPB test methods are already standardized as ISO 1209-2:2007 [19] and JIS K 7221-2:2006 [20], and sample configurations are determined in these standards. Therefore, bending properties have been conventionally examined based on these standardized methods. In contrast, the investigations on the effects of testing methods and sample configurations have often been neglected. If these effects are revealed, the accuracy in characterizing the bending properties must be improved and the frequency of conducting static bending tests will increase. Additionally, there are no examples where the CB test was adopted for the characterization of the bending properties of XPS; therefore, it is valuable to perform a CB test when using XPS. It is valuable to compare the results obtained from the different tests using various sample configurations for the accurate characterization of the bending properties of XPS.

In this study, TPB, FPB, and CB tests were performed using XPS samples at various span/depth ratios. The effects of the chosen testing methods and sample configurations were examined through the use of analysis of variance (ANOVA), and the most feasible method for characterizing the bending properties of these samples was investigated using statistical analyses.

## 2. Theoretical Background

Cellular plastics are often used in shock-absorbing layers under compressive loading [31,32,33]; therefore, compression tests are more frequently performed than other mechanical tests. However, to obtain the Young’s modulus via a compression test, further apparatus, such as a compressometer, is required to measure the strain along the loading direction. Otherwise, the strain must be measured via some optical method, such as electronic spec pattern interferometry (ESPI) and digital image correlation (DIC). In contrast, as described above, a static bending test can be performed easily. In particular, it is more advantageous than the compression test in that the abovementioned apparatus and technique are not required to obtain the Young’s modulus.

Figure 1a–c illustrate diagrams of the TPB, FPB, and CB tests, respectively.

The TPB test method is standardized by the International Organization for Standardization and Japanese Industrial Standards as ISO 1209-02:2007 [19] and JIS K 7221-2:2006 [20], respectively, for characterizing the bending properties of rigid cellular plastics. The load applied to the midspan *P* and deflection at midspan *δ*_M_ can be obtained, as shown in Figure 1a. From EBT, the Young’s modulus in the length direction of sample *E_x_* is obtained as follows:(1)$$E_{x}=\frac{3L^{3}}{4BH^{3}}\cdot \frac{\Delta P}{\Delta \delta_{M}}$$
where *L* is the distance between the supports, *B* and *H* are the width and depth of the sample, respectively, and Δ*P*/Δ*δ*_M_ represents the slope of the straight line drawn along the initial linear segment in the *P*-*δ*_M_ relationship. The bending stress at the outer surface of the midspan *σ_x_* is also derived from EBT as follows:(2)$$\sigma_{x}=\frac{3PL}{2BH^{2}}$$

Therefore, the proportional limit stress *Y_x_* is determined as
(3)$$Y_{x}=\frac{3P_{NL}L}{2BH^{2}}$$
where *P*_NL_ is the load where the aforementioned straight line deviates from the *P*-*δ*_M_ curve. The method used to determine *P*_NL_ values is described below. The bending strength *F_x_* can be obtained by substituting the maximum load *P*_max_ into *P* in Equation (2). However, when the deflection is extremely large, the direction of the reaction force from the supporting point is inclined rather than positioned upward; therefore, it is often difficult to determine the load–deflection relationship and bending strength using EBT. For carbon-fiber-reinforced plastics (CFRPs), the methods used to calculate the bending strength are standardized with consideration of the effect of large deflection as ASTM D 790-17 [34] and JIS K 7074-1988 [35], in which the equations used are different. The equations determined in these standards were compared with those based on EBT, as follows:(4)$$F_{x}=\left\{\begin{array}{c} \frac{3P_{\max}L}{2BH^{2}}\;\left(EBT\right)\\ \frac{3P_{max}L}{2BH^{2}}\left[1+6{\left(\frac{\delta_{Mb}}{L}\right)}^{2}-4\left(\frac{\delta_{Mb}}{L}\right)\left(\frac{H}{L}\right)\right]\;(ASTM\;D790–17)\\ \frac{3P_{max}L}{2BH^{2}}\left[1+4{\left(\frac{\delta_{Mb}}{L}\right)}^{2}\right]\;\left(JIS\;K\;7074–1988\right) \end{array}\right.$$
where *δ*_Mb_ = *δ*_M_ at *P* = *P*_max_.

Although the FPB test used for rigid cellular plastics is not determined in major standards, it can be performed as easily as the TPB test; therefore, FPB test methods for various materials have been determined in the case of several major standards. As shown in Figure 1b, in the FPB test, two loads of *P*/2 are symmetrically applied to the trisectional points, and the deflection at the loading point is defined as *δ*_L_. In the case of FPB test methods for CFRP standardized in ASTM D6272-17 [36] and JIS K 7074-1988 [35], the deflection is measured at the midspan using equipment such as a linear variable differential transformer (LVDT). However, for lightweight materials, such as XPS, deflection cannot be measured accurately because the reaction force from the LVDT is often large, affecting the bending deformation of the sample. Instead of using an LVDT, *δ*_M_ is calculated from *δ*_L_, based on EBT, as follows:(5)$$\delta_{M}=\frac{23}{20}\delta_{L}$$

In this study, the distance between the loading noses was one-third that of the support span, as determined in ASTM D6272-17 [36] and JIS K 7074-1988 [35]. Therefore, *E_x_* and *σ_x_* are obtained from EBT as follows:(6)$$E_{x}=\frac{23L^{3}}{108BH^{3}}\cdot \frac{\Delta P}{\Delta \delta_{M}}$$
and
(7)$$\sigma_{x}=\frac{PL}{BH^{2}}$$

In the FPB test, *Y_x_* is determined using Equation (7) as follows:(8)$$Y_{x}=\frac{P_{NL}L}{BH^{2}}$$

In ASTM D 6272-17 [36] and JIS K 7074-1988 [35], the equations for FPB are also standardized using different equations, and they are demonstrated with the equation derived from EBT as follows:(9)$$F_{x}=\left\{\begin{array}{c} \frac{P_{max}L}{BH^{2}}\;\left(EBT\right)\\ \frac{P_{max}L}{BH^{2}}\left[1+4.70{\left(\frac{\delta_{Mb}}{L}\right)}^{2}-7.04\left(\frac{\delta_{Mb}}{L}\right)\left(\frac{H}{L}\right)\right]\;(ASTM\;D6272–17)\\ \frac{P_{max}L}{BH^{2}}\left[1+\frac{4644}{529}{\left(\frac{\delta_{Mb}}{L}\right)}^{2}-\frac{162}{23}\left(\frac{\delta_{Mb}}{L}\right)\left(\frac{H}{L}\right)\right]\;\left(JIS\;K\;7074–1988\right) \end{array}\right.$$

In the CB test shown in Figure 1c, the angle between the vertical axis and length direction at the end of the sample is defined as *α*. In addition, the displacements at the loading point are defined as *x*, whereas the curvature at the midspan is defined as *κ*_Μ_. According to elastica theory, *x*/*L* = 0–0.0676, 0.0676–0.259, and 0.259–0.543 in the ranges of *α* = 0–30°, 30–60°, and 60–90°, respectively, and the *δ*_M_/*L*-*x*/*L* and *κ*_M_*L*-*x*/*L* relationships under the hinged ends are approximated as follows [30]:(10)$$\frac{\delta_{M}}{L}=\left\{\begin{array}{c} 0.62390{\left(\frac{x}{L}\right)}^{0.49745}\;\left(0\leq x/L\leq 0.0676\right)\\ 0.54880{\left(\frac{x}{L}\right)}^{0.45036}\;\left(0.0676<x/L\leq 0.259\right)\\ 0.47334{\left(\frac{x}{L}\right)}^{0.33983}\;\left(0.259<x/L\leq 0.543\right) \end{array}\right.$$
and
(11)$$\kappa_{M}L=\left\{\begin{array}{c} 6.3625{\left(\frac{x}{L}\right)}^{0.50153}\;\left(0\leq x/L\leq 0.0676\right)\\ 6.8713{\left(\frac{x}{L}\right)}^{0.52978}\;\left(0.0676<x/L\leq 0.259\right)\\ 7.5113{\left(\frac{x}{L}\right)}^{0.59634}\;\left(0.259<x/L\leq 0.543\right) \end{array}\right.$$

From the relationships derived using Equation (10), *σ_x_* can be derived as
(12)$$\sigma_{x}=\frac{6P\delta_{M}}{BH^{2}}$$
when the longitudinal strain at the midspan surface is defined as the bending strain *ε_x_*, it is derived using *κ*_M_, as follows:(13)$$\varepsilon_{x}=\frac{\kappa_{M}H}{2}$$

*E_x_*, *Y_x_*, and *F_x_* can be obtained using the *σ_x_*-*ε_x_* relationship derived from Equations (12) and (13).

## 3. Materials and Methods

### 3.1. Materials

The test samples used in this study were cut from a commercially sold XPS panel (STYROFOAM B2; Dupont Styro Corporation, Tokyo, Japan). The initial dimensions of the sold panel were 1820, 910, and 25 mm in length, width, and thickness, respectively. The length, width, and thickness directions of the panel are defined as the L-, T-, and Z-axes, respectively. The panel was initially cut using a circular saw and finished to the final dimensions of the sample using a heat wire. The L-, T-, and Z-axes of the panel coincided with the length, depth, and width directions of the sample, respectively; therefore, sample width *B* was 25 mm. In contrast, the sample depths, H, were cut to 20 mm and 10 mm. For the sample with *H* = 20 mm, the length varied from 300 mm to 700 mm at intervals of 100 mm; in contrast, it varied from 500 mm to 700 mm at intervals of 50 mm for the sample with *H* = 10 mm. The distance between supports *L* varied from 100 to 500 mm at intervals of 100 mm and from 300 to 500 mm at intervals of 50 mm in the samples with *H* = 20 and 10 mm, respectively. Therefore, the *L*/*H* value varied from 5 to 50 at intervals of 5. The sample was cut such that the length of the overhung portion was 100 mm. Five samples were used for each test. The density of the sample was 25.5 ± 0.3 kg/m^3^. In the TPB test, the load was applied at the midspan (ASTM D 790-17 [34] and JIS K 7074-1988 [35]). In the FPB test, the distance between the loading noses was one-third that of the support span (ASTM D6272-17 [36] and JIS K 7074-1988 [37]). The CB test was performed according to the method described in Yoshihara et al. [30].

### 3.2. TPB, FPB, and CB Tests

TPB, FPB, and CB tests were performed, and *E_x_*, *Y_x_*, and *F_x_* values were obtained by varying the *L*/*H* value.

Figure 2 shows photographs of the TPB, FPB, and CB tests conducted in this study. In the TPB and FPB tests, as shown in Figure 2a and Figure 2b, respectively, the sample was set on a pair of supports with a radius of 10 mm. A load, *P*, was applied using a loading nose with a radius of 10 mm. In the CB test, when using a sample with *H* = 20 mm, cylindrical attachments were set on both ends of the sample to enhance bending deformation and a load was applied via the cylindrical attachment using a steel plate, as shown in Figure 2c, to enhance the rotation at the sample ends. However, when using the sample with *H* = 10 mm, bending deformation was induced by the weight of the attachment set on the top of the sample prior to the application of the axial load, and *E_x_*, *Y_x_*, and *F_x_* could not be appropriately determined using the *σ_x_*-*ε_x_* relationship. To solve this issue, the CB test was performed without the use of the top attachment, as shown in Figure 2d. Even under this condition, rotation was easily induced at the sample ends, and the bending properties were appropriately determined.

The crosshead speed $\dot{x}$ was derived from the following relationship [30,37]:(14)$$\dot{x}=\left\{\begin{array}{c} \frac{\dot{\varepsilon_{x}}L^{2}}{6H}\;\left(TPB\right)\\ \frac{23\dot{\varepsilon_{x}}L^{2}}{108H}\;(FPB)\\ L{\left(\frac{0.31434\dot{\varepsilon_{x}}L}{H}\right)}^{1.9939}\;\left(CB\right) \end{array}\right.$$

Similar to the results of a previous study, the strain rate at the midspan $\dot{\varepsilon_{x}}$ was determined to be 0.06/s [30]. Table 1 lists the crosshead speeds adopted under each test condition.

As described above, the *P*-*δ*_M_ curve was obtained in the TPB and FPB tests, and Δ*P*/Δ*δ*_M_ and *P*_NL_ were determined from a straight line drawn along the initial linear segment of the *P*-*δ*_M_ curve. The *P*_NL_ value, defined as the load at the deviation of linearity, was determined as the load where the half-thickness of the plotter trace of the *P*-*δ*_M_ curve deviated from the aforementioned straight line [38]. *E_x_* was determined by substituting Δ*P*/Δ*δ*_M_ into Equations (1) and (6) in the TPB and FPB tests, respectively. Additionally, *Y_x_* was determined by substituting *P*_NL_ into Equations (3) and (8) for the TPB and FPB tests, respectively. In contrast, *P*_max_ was obtained from the maximum load, and *F_x_* was determined by substituting *P*_max_ into Equations (4) and (9) for the TPB and FPB tests, respectively. The effects of the equations obtained from EBT, ASTM standard, and JIS standard were examined by comparing the results, and the superiorities of the equations were investigated. In the CB test, *E_x_*, *Y_x_*, and *F_x_* were directly determined from the *σ_x_*-*ε_x_* relationship obtained from Equations (12) and (13).

## 4. Results and Discussion

### 4.1. Experimental Results Obtained from the TPB, FPB, and CB Tests

In the TPB and FPB tests, the *σ_x_*-*ε_x_* relationship can be determined based on the results of EBT; therefore, the *P*-*δ*_M_ relationships were transformed into *σ_x_*-*ε_x_* relationships, and the effects of the test methods and sample configurations were compared using the *σ_x_*-*ε_x_* relationships. In the TPB test, *ε_x_* is derived as follows:(15)$$\varepsilon_{x}=\frac{{6B\delta }_{M}}{H^{2}}$$

In the FPB test, *ε_x_* is derived as follows:(16)$$\varepsilon_{x}=\frac{{108B\delta }_{M}}{23H^{2}}$$

Figure 3 illustrates comparisons of the representative *σ_x_*-*ε_x_* relationships calculated using Equations (2) and (15) (TPB tests), Equations (7) and (16) (FPB tests), and Equations (12) and (13) (CB tests). In the TPB and FPB tests, when the *L*/*H* value is sufficiently low, bending failure is often induced in an unstable manner when *σ_x_* reaches *σ*_max_. In contrast, when the *L*/*H* value increases, this tendency is not enhanced; however, *σ_x_* gradually decreases after it reaches *σ*_max_. Additionally, the *σ*_max_ value obtained from EBT decreases as *L*/*H* increases, particularly in the FPB test results. In contrast, in the CB tests, the *σ_x_*-*ε_x_* relationships were found to be similar to one another, except for those with *L*/*H* values of 5 and 10, and bending failure was induced immediately after *σ_x_* reached *σ*_max_.

In the standards for bending tests of CFRP (ASTM D790-17 [34] and ASTM D6272-17 [36]), the methods determined in these standards are not applicable when *ε_x_* exceeds 0.05, and other mechanical properties, such as tensile strength, may be more relevant for characterizing different materials. The results shown in Figure 3 indicate that *ε_x_* at the maximum *σ_x_* often exceeds 0.05 in TPB and FPB tests on samples with *L*/*H* values lower than 25. In the standards for the bending tests of cellular rid plastics (ISO 1209-2:2007 [19] and JIS K 7221-2:2006 [20]), the *L*/*H* value is determined to be 12-16. However, to determine the bending strength of XPS under *ε_x_* < 0.05, the *L*/*H* value should be greater than 16. In contrast, in the CB test, a plateau portion in the *σ_x_*-*ε_x_* relationship was found within the 0.05 *ε_x_* limit when using the samples with *L*/*H* values higher than 15.

The results shown in Figure 4 demonstrate the dependence of *E_x_* on *L*/*H* in the TPB, FPB, and CB tests. In the TPB and FPB tests, the ratio of the deflection caused by the shearing force to that by the bending moment increases as the *L*/*H* value decreases. Since Equations (1) and (6) used for calculating *E_x_* values in the TPB and FPB tests, respectively, they are based on EBT, where the effect of the deflection caused by the shearing force is ignored. Therefore, the *E_x_* values decrease as the *L*/*H* value decreases. The low values of *E_x_* at *L*/*H* = 5 in the TPB and FPB tests were due to the effect of deflection by the shearing force. However, although the results of previous studies on TPB tests performed using solid wood [39,40] indicate that the dependence of *E_x_* on *L*/*H* decreases as the *L*/*H* value increases, the *E_x_* values continuously increase in the *L*/*H* ranges of 5–50 and 5–40 in the TPB and FPB tests, respectively. In these tests, a slight slippage was induced between the sample and support even when the load was not sufficiently large. Therefore, owing to the slippage, which was induced during the flexural loading, the sample is supported at the points outward than those at the initial points; therefore, the sample behaves as if the distance between the supports were larger than the actual distance. The degree of slippage increased as the *L*/*H* value increased; therefore, it can be concluded that the *E_x_* value increases as the *L*/*H* value increases. In the CB test, the *E_x_* values at *L*/*H* = 5 and 10 were lower than those at *L*/*H* ≥ 15. When the *L*/*H* value is low, material nonlinearity is induced, and the stiffness of the sample is usually reduced due to the occurrence of material nonlinearity. In contrast, when the CB test is performed using a sample with an *L*/*H* value higher than 15, the dependence of *E_x_* on *L*/*H* is effectively reduced. Therefore, when a sample with an *L*/*H* value lower than 10 was not used, it was more advantageous to use the CB test than the TPB and FPB tests because *E_x_* can be obtained without slippage between the sample and support. However, the *E_x_* values obtained from the CB tests were often lower than those obtained from the TPB and FPB tests. Therefore, further research is required in order to elucidate the reasons for these differences.

Figure 5 shows the dependence of *Y_x_* on *L*/*H* in the TPB, FPB, and CB tests. The fluctuation in the *Y_x_* value is more obvious than those in the *E_x_* and *F_x_* values. The method for determining the *Y_x_* value described above is more error-prone; therefore, the variation in the *Y_x_* value is greater than those in the *E_x_* and *F_x_* values. Our statistical analyses revealed no significant differences in the results obtained from the TPB tests due to the large variation. Despite the fluctuation, significant differences were frequently observed in the results obtained from the FPB and CB tests. The details of the results obtained from the statistical analyses are demonstrated below.

Figure 6 shows the dependence of *F_x_* on *L*/*H* in the TPB, FPB, and CB tests. In the TPB test, when *F_x_* is calculated based on EBT, it decreases as the *L*/*H* value increases. However, when *F_x_* is calculated using the equations standardized in ASTM D790-17 [34] and JIS K 7074-1988 [35], the dependence of *F_x_* on *L*/*H* is effectively reduced. No dependence was observed in the results obtained from the CB tests. In contrast, in the FPB test, the dependence was found to be significant, and the equations determined in ASTM D6272-17 [36] and JIS K 7074-1988 [38] were not effective in reducing the dependence. 

In the following section, we discuss the results of our statistical analyses of the samples’ bending properties and the validity of the TPB, FPB, and CB tests.

### 4.2. Statistical Analyses of the Testing Results

As with the statistical analysis, ANOVA (Tukey’s tests) was performed to assess the difference between the *E_x_*, *Y_x_*, and *F_x_* values for different *L*/*H* values. Using the results obtained from the ANOVA, the effects of *L*/*H* and the test methods used are discussed below from a statistical perspective. EZR, developed by the Jichi Medical University Saitama Medical Center, was used to perform the ANOVA [41].

Table 2 lists the results of the ANOVA for *E_x_* corresponding to *L*/*H*. In the TPB and FPB tests, the differences among the averages of *E_x_* were not found in the *L*/*H* value range of 25 to 50. In contrast, when 5 ≤ *L*/*H* ≤ 20, the *E_x_* values were lower than those in the range of *L*/*H* values higher than 25 because the deflection due to the shearing force cannot be ignored. Although the *L*/*H* was determined to be 15 for the TPB test in ISO 1209-2:2007 [19] and JIS K 7221-2:2006 [20], the results of the statistical analyses suggest that it is preferable to perform TPB tests under *L*/*H* values higher than the standardized value. In contrast, in the CB test, differences among the averages of *E_x_* were not found in the *L*/*H* value range of 15 to 50, which is wider than the range found in the TPB and FPB tests. Therefore, when considering the range of *L*/*H* alone, the CB test is superior to the TPB and FPB tests.

Table 3 lists the results of the ANOVA for *Y_x_* corresponding to *L*/*H*. As described above, there were no significant differences between the *Y_x_* values obtained from the TPB tests. In contrast, in the FPB and CB tests, the *L*/*H* range where the differences are not significant is restricted to 20 ≤ *L*/*H* ≤ 50 and 15 ≤ *L*/*H* ≤ 50, respectively.

Table 4 lists the results of the ANOVA for *F_x_* corresponding to *L*/*H*. In the TPB tests, the range of *L*/*H* values where the differences among the averages are not significant is restricted to 15 ≤ *L*/*H* ≤ 50 when using the equations based on EBT, but the significant differences are reduced when *F_x_* is calculated using the equations determined in ASTM D 790-17 [34] and JIS K 7074-1988 [35]. In contrast, the differences cannot be reduced sufficiently in the FPB tests, even when ASTM- and JIS-standardized equations are used. In the CB test, no significant differences were observed between the average *F_x_*. The results of the statistical analyses show that the TPB and CB tests are more advantageous than the FPB test.

Table 5 lists the *E_x_*, *Y_x_*, and *F_x_* values obtained by averaging those in the *L*/*H* range, where the significance level is higher than 0.05. The *E_x_*, *Y_x_*, and *F_x_* values obtained from the TPB and FPB tests were similar. In contrast, the *E_x_* and *F_x_* values obtained from the CB tests were lower than those obtained from the TPB and FPB tests, whereas the *Y_x_* values obtained from the CB tests were close to those obtained from the TPB and FPB tests. These tendencies are often discrepant from those obtained using CFRP [24,25,26], solid wood [27], plywood and MDF [28,29], and cardboard [30]. Cellular structures in XPS may affect their properties. Further research, including microscopic observations, is thus required to elucidate the discrepancies between the results.

Summarizing the results obtained in this study, performing TPB and CB tests under the span/depth ratios of 25–50 and 15–50, respectively, is promising in terms of determining the bending properties of XPS (Young’s modulus, proportional limit stress, and bending strength) used in this study. Further research should also be conducted to establish the appropriate testing method and sample configuration by conducting static bending tests using various XPS samples.

## 5. Conclusions

In this study, three-point bending (TPB), four-point bending (FPB), and compression bending (CB) tests were performed on extruded polystyrene (XPS) samples to determine their bending properties. The dependence of the properties on the span/depth ratio was statistically analyzed (Tukey’s tests), and the applicability of the three test methods was examined. Through our analyses, the following results were obtained:(1)In the TPB and FPB tests, Young’s modulus increased continuously as the span/depth ratio increased because of the slippage between the sample and supporting point, as well as the effect of deflection caused by the shearing force. In the CB test, the Young’s modulus was lower in the small span/depth ratio range where material nonlinearity was induced by axial loading prior to bending deformation. Except for this span/depth ratio range, Young’s modulus could be obtained while reducing the effect of the sample configuration.(2)The effect of the span/depth ratio on the proportional limit stress was not observed in the TPB test results; in contrast, the effect was found in the results obtained from the FPB and CB tests.(3)To determine the samples’ bending strength by using the TPB test, the methods for correcting the large defection determined in ASTM D 790-17 and JIS K 7074-1988 were effective; the correction methods for the FPB test determined in ASTM 6272-17 and JIS K 7074-1988 were not sufficient, however. In contrast, the effect of the span/depth ratio was not significant in the CB test results.(4)From the experimental results, the use of TPB and CB tests is recommended over the FPB test to determine the bending properties of XPS samples. However, the properties determined via the TPB and CB tests were different. Therefore, further research should be conducted to explore the source of these differences in more detail.

## Figures and Tables

**Figure 1 polymers-16-01857-f001:**
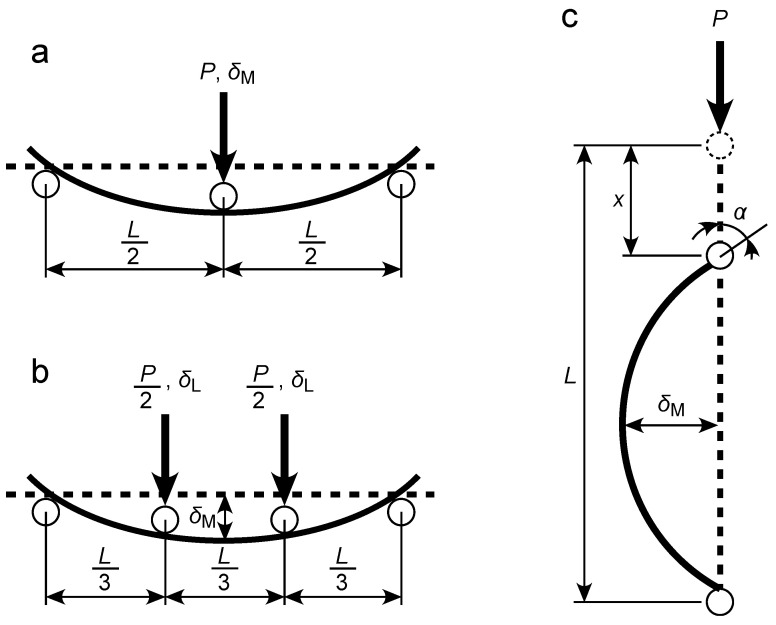
Diagrams of the TPB (**a**), FPB (**b**), and CB (**c**) tests.

**Figure 2 polymers-16-01857-f002:**
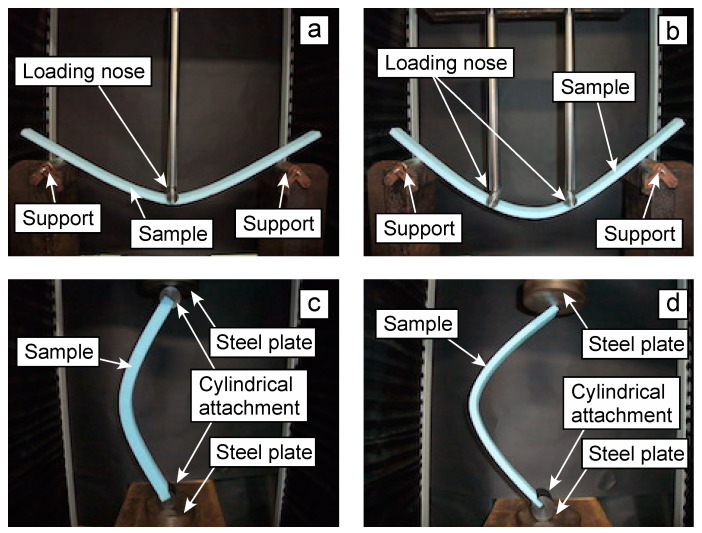
Photographs of the TPB test (**a**), FPB test (**b**), CB test using a sample with *H* = 20 mm (**c**), and CB test using a sample with *H* = 10 mm (**d**).

**Figure 3 polymers-16-01857-f003:**
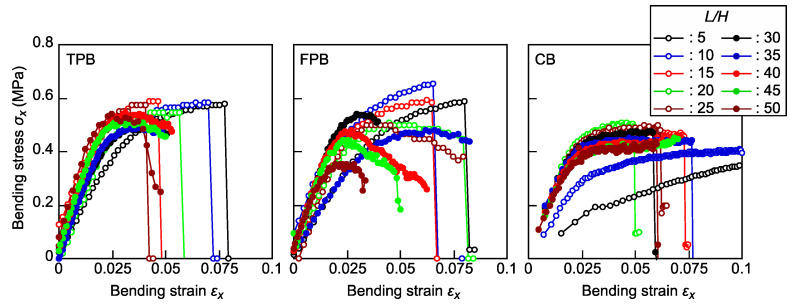
*σ_x_*-*ε_x_* relationships obtained by varying the *L*/*H* values in the TPB, FPB, and CB tests.

**Figure 4 polymers-16-01857-f004:**
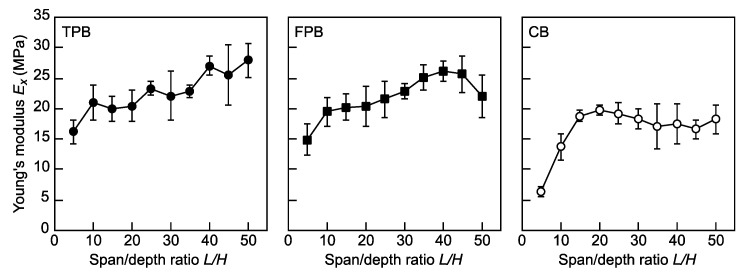
Dependence of *E_x_* on *L*/*H* in the TPB, FPB, and CB tests. Data = average ± standard deviations.

**Figure 5 polymers-16-01857-f005:**
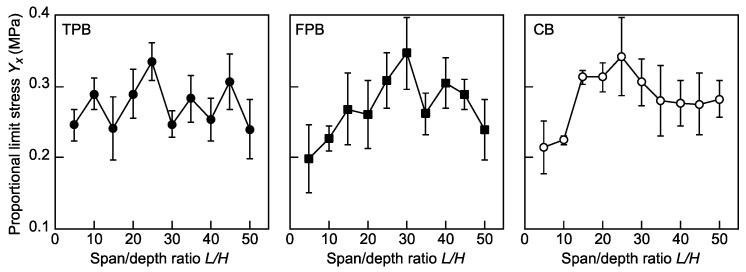
Dependence of *Y_x_* on *L*/*H* in the TPB, FPB, and CB tests. Data = average ± standard deviations.

**Figure 6 polymers-16-01857-f006:**
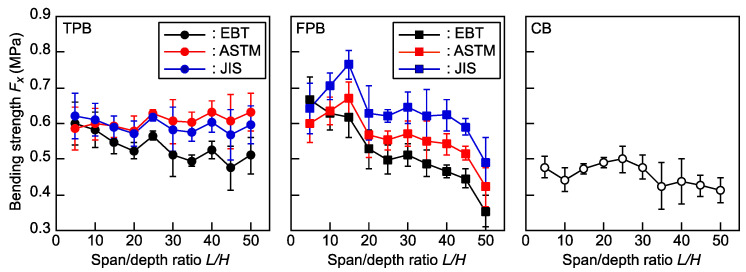
Dependence of *F_x_* on *L*/*H* ratio in the TPB, FPB, and CB tests. Data = average ± standard deviations. EBT = elementary beam theory, ASTM = ASTM D790-17 and ASTM D6272-17 in the TPB and FPB tests, respectively, and JIS = JIS K 7074-1988.

**Table 1 polymers-16-01857-t001:** Crosshead speed in the TPB, FPB, and CB tests.

*H* (mm)	20					10				
*L* (mm)	100	200	300	400	500	300	350	400	450	500
*L*/*H*	5	10	15	20	25	30	35	40	45	50
TPB	5.0	20.0	45.0	80.0	125	90.0	123	160	203	250
FPB	6.39	25.6	57.5	102	160	115	156	205	259	319
CB	0.902	7.19	24.2	57.3	112	96.4	153	228	324	445

Unit of crosshead speed = mm/min.

**Table 2 polymers-16-01857-t002:** ANOVA (Tukey’s test) results for *Ex* corresponding to *L*/*H*.

*L*/*H*	5	10	15	20	25	30	35	40	45	50
TPB	A	AB	AC	ACD	BCD**E**	BCD**E**	BCD**E**	**E**	BD**E**	**E**
FPB	A	AB	ABC	B	B**D**	B**D**	C**D**	**D**	C**D**	C**D**
CB		A	**B**	**B**	**B**	**B**	A**B**	A**B**	**B**	**B**

Significant differences are not found among the samples with the same letters in the horizontal row. The same bold letters in the horizontal row represent the widest range of *L*/*H* where significant differences among the samples are not found.

**Table 3 polymers-16-01857-t003:** ANOVA (Tukey’s test) results for *Yx* corresponding to the different *L*/*H*.

*L*/*H*	5	10	15	20	25	30	35	40	45	50
TPB	**A**	**A**	**A**	**A**	**A**	**A**	**A**	**A**	**A**	**A**
FPB	A	AB	AC	A**D**	BC**D**	C**D**	AC**D**	BC**D**	AC**D**	A**D**
CB	A	AB	**C**	**C**	**C**	B**C**	A**C**	AB**C**	AB**C**	AB**C**

Significant differences are not found among the samples with the same letters in the horizontal row. The same bold letters in the horizontal row represent the widest range of *L*/*H* where significant differences among the samples are not found.

**Table 4 polymers-16-01857-t004:** ANOVA (Tukey’s tests) results for *Fx* corresponding to the different *L*/*H*.

*L*/*H*	5	10	15	20	25	30	35	40	45	50
TPB, EBT	A	AB	AB**C**	AB**C**	AB**C**	B**C**	B**C**	AB**C**	**C**	AB**C**
TPB, ASTM	**A**	**A**	**A**	**A**	**A**	**A**	**A**	**A**	**A**	**A**
TPB, JIS	**A**	**A**	**A**	**A**	**A**	**A**	**A**	**A**	**A**	**A**
FPB EBT	A	A	AB	B**C**	**C**	**C**	**C**	**C**	**C**D	D
FPB, ASTM	A	AB	A	A**C**	A**C**	A**C**	B**C**	B**C**	**C**D	D
FPB, JIS	A	AB	A	A**C**	A**C**	A**C**	B**C**	B**C**	**C**D	**C**
CB	**A**	**A**	**A**	**A**	**A**	**A**	**A**	**A**	**A**	**A**

Significant differences are not found among the samples with the same letters in the horizontal row. The same bold letters in the horizontal row represent the widest range of *L*/*H* where the significant differences among the samples are not found. EBT = elementary beam theory, ASTM = ASTM D790-17 and ASTM D6272-17 in the TPB and FPB tests, respectively, and JIS = JIS K 7074-1988.

**Table 5 polymers-16-01857-t005:** *E_x_*, *Y_x_*, and *F_x_* in the *L*/*H* range, where the significance level exceeded 0.05.

Method	*E_x_* (MPa)	*Y_x_* (MPa)	*F_x_* EBT (MPa)	*F_x_* ASTM (MPa)	*F_x_* JIS (MPa)
TPB	24.8 ± 3.5	0.273 ± 0.049	0.520 ± 0.045	0.606 ± 0.047	0.593 ± 0.046
FPB	23.9 ± 3.0	0.287 ± 0.049	0.490 ± 0.043	0.551 ± 0.042	0.602 ± 0.070
CB	18.1 ± 2.5	0.298 ± 0.040	0.457 ± 0.046	-	-

Data = average ± standard deviations. EBT = elementary beam theory, ASTM = ASTM D790-17 and ASTM D6272-17 in the TPB and FPB tests, respectively, and JIS = JIS K 7074-1988.

## Data Availability

Data are contained within the article.

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
