# Peer review of "Effects of Testing Methods and Sample Configuration on the Flexural Properties of Extruded Polystyrene"

_polymers, 2024, doi:10.3390/polym16131857_

Round 1
Reviewer 1 Report
Comments and Suggestions for Authors
The objective of this work is to investigate the influence of sample configuration on bending properties of extruded polystyrene.
In my opinion the article should be revised since some concepts are confusing and need to be clarified. Some suggestions in order to improve the article are listed below.
Pag. 3. Line 84: Please, correct: “…where L is the distance between the spans…” L or “span”, is the distance between supports.
Pag. 3. Lines 110-111. Equation 5. Please specify the test configuration used in the four points bending test: one third of support span or one half of support span (ASTM D 6272) This should also be made clear in page 5, lines 138-144, when describing the test configuration.
Pag. 5. Line 150. “A lateral load, P, was applied…” Please, consider: “A vertical load was applied…” as the term “lateral” could be confusing.
The quality of figure 2 should be improved. Figures have poor definition in the evaluated pdf.
Page 6. Line 174. Please, improve the wording “The PNL value was determined using the load for deviation from…”
Page 6. Line 175. Please clarify the following sentence. It is not clear what you mean. “…at half the thickness of the plotted curve [32].”
Page 7. Lines 211-212. Please, rewrite in order to clarify the concept: “…in the TPB and FPB tests, Ex usually decreases the deflection induced by the shearing force, which is enhanced as the L/H value decreases”.
Page 7. Line 219: Please, correct “…were supported by spans…” Span length or span is the distance between supports.
Page 7. Line 219. “Therefore, owing to this slippage, the sample deformed as if it were supported by spans with a distance larger than the actual one”. The span does not change during the test. The distance between the loading axis and the supports remains constant. The reasoning behind this statement is unclear.
Page 7. Line 228. Please correct “stably”
Page 7. Line 228. “...between the sample and span” Please, correct “span”
Page 9. Line 270-271. “…in the horizontal column” Do you mean “row”? Please, correct.
Pages 9 and 10. The meaning of the different letters in tables 2, 3 and 4 should be explained in depth. The explanation is confusing.
Comments on the Quality of English LanguagePlease, check the quality of the English language. For example, the use of “span” seems erroneous in some contexts in the article: “span length” or “span” is the distance between supports.
Page 6. Line 174. Please, improve the wording in: “The PNL value was determined using the load for deviation from…”
Page 7. Line 228. Please correct “stably”
Page 9. Line 270-271. “…in the horizontal column” Do you mean “row”? Please, correct.
Reviewer 2 Report
Comments and Suggestions for Authors
This manuscript reports on the measurement of the bending properties of extruded polystyrene (XPS) using three established methods: three-point bending (TPB), four-point bending (FPB), and compression bending (CB). The primary concern with this manuscript is its lack of novelty. These three tests are already standardized, and the manuscript does not introduce any new testing methods or innovative ideas. Consequently, I recommend that the manuscript be rejected.
Comments on the Quality of English Language-
Reviewer 3 Report
Comments and Suggestions for Authors
The manuscript Polymers-3032203 entitled “The Effect of Sample Configuration on the Measurement of the Bending Properties of Extruded Polystyrene Using Three-Point, Four-Point, and Compression Bending Tests” is well structured and well written. However, the authors are asked to respond to all of the following comments:
1. The title of a scientific article should be short and concise. I proposes to use the following title “Effect of sample configuration on the flexural properties of extruded polystyrene”;
2. The illustrations in Figure 1 are very explicit and the Figure 2 with photographs therefore become unnecessary;
3. What was the reason for performing the bending test using two different standards: an American standard (ASTM) and a Japanese standard (JIS)?;
4. Typically, cellular plastics are tested in compression using a load perpendicular to the sample surface. The authors did not explain why they used the test with the sample shown in Figure 1;
5. In order to provide easy, I recommend to join the two sections 3.Results and 4. Discussion;
6. The ANOVA (Tukey’s tests) is mentioned with relevant results but without explanation;
7. The references are not reported uniformly. Some of them should be corrected;
8. Other remarks:
ü The abbreviation “EBT” should be defined in the text;
ü Section 3.Results on page 7, line 205, page line : Replace the word “solid” by “rid”
ü The Section 6.Nomenclature on page 11 should be removed since all the factors (parameters) are explained in the text.
Comments on the Quality of English Languagevery good english
Reviewer 4 Report
Comments and Suggestions for Authors
The manuscript title “The Effect of Sample Configuration on the Measurement of the 2 Bending Properties of Extruded Polystyrene Using Three-Point, 3 Four-Point, and Compression Bending Tests” have a huge potential in polymer mechanical testing analysis. However, amendment required as follows:
- No significant outcome and findings mentioned in abstract. The abstract is too general and difficult to identify novelty of this study.
- Introduction is too general, and the research gap of this article is unclear. The author suggested to highlight the novelty findings and explain more on the background study in introduction parts
- If the testing performed based on ISO standard hence Fig 1 is not required
- The initial dimension of the samples 1820,910 and 25mm. where that these dimensions coming from? Based on ISO standard or prior research?
- Fig 2 is unnecessary unless the author planned to provided quality images of deformation process (DIC method).
- Fig 5, the data is fluctuate. Is there any explanation on this?the error bar also quite high considering this samples experience any slip during the testing?
- The comparision made in this article is too general and been reported in prior study. Hence what is the major highlight of this article? Any micrograpgh images on behaviour or mechanism difference will be helpful.
Round 2
Reviewer 2 Report
Comments and Suggestions for Authors
The study aims to compare TPB, FPB, and CB tests on XPS samples with various span/depth ratios, utilizing analysis of variance (ANOVA) to identify the most feasible method for characterizing the bending properties of XPS. This comparative approach is intended to enhance the accuracy of mechanical property measurements and encourage the increased use of static bending tests for XPS.
I believe the manuscript is suitable for publication.
Reviewer 4 Report
Comments and Suggestions for Authors
accepted